# Computer-Aided Diagnosis of Multiple Sclerosis Using a Support Vector Machine and Optical Coherence Tomography Features

**DOI:** 10.3390/s19235323

**Published:** 2019-12-03

**Authors:** Carlo Cavaliere, Elisa Vilades, Mª C. Alonso-Rodríguez, María Jesús Rodrigo, Luis Emilio Pablo, Juan Manuel Miguel, Elena López-Guillén, Eva Mª Sánchez Morla, Luciano Boquete, Elena Garcia-Martin

**Affiliations:** 1Biomedical Engineering Group, Department of Electronics, University of Alcalá, 28801 Alcalá de Henares, Spain; carlo.cavaliere@uah.es (C.C.); jmanuel.miguel@uah.es (J.M.M.); elena.lopezg@uah.es (E.L.-G.); 2Department of Ophthalmology, Miguel Servet University Hospital, 50009 Zaragoza, Spain; elisavilades@hotmail.com (E.V.); lpablo@unizar.es (L.E.P.); 3Aragon Institute for Health Research (IIS Aragon), Miguel Servet Ophthalmology Innovation and Research Group (GIMSO), University of Zaragoza, 50009 Zaragoza, Spain; 4Department of Physics and Mathematics, University of Alcalá, 28801 Alcalá de Henares, Spain; mconcepcion.alonso@uah.es; 5RETICS-Oftared: Thematic Networks for Co-operative Research in Health for Ocular Diseases, 28040 Madrid, Spain; 6Department of Psychiatry, 12 Octubre University Hospital Research Institute (i+12), 28041 Madrid, Spain; emsmorla@gmail.com; 7Faculty of Medicine, Complutense University of Madrid, 28040 Madrid, Spain; 8CIBERSAM: Biomedical Research Networking Centre in Mental Health, 28029 Madrid, Spain

**Keywords:** multiple sclerosis, optical coherence tomography, support vector machine, confusion matrix

## Abstract

The purpose of this paper is to evaluate the feasibility of diagnosing multiple sclerosis (MS) using optical coherence tomography (OCT) data and a support vector machine (SVM) as an automatic classifier. Forty-eight MS patients without symptoms of optic neuritis and forty-eight healthy control subjects were selected. Swept-source optical coherence tomography (SS-OCT) was performed using a DRI (deep-range imaging) Triton OCT device (Topcon Corp., Tokyo, Japan). Mean values (right and left eye) for macular thickness (retinal and choroidal layers) and peripapillary area (retinal nerve fibre layer, retinal, ganglion cell layer—GCL, and choroidal layers) were compared between both groups. Based on the analysis of the area under the receiver operator characteristic curve (AUC), the 3 variables with the greatest discriminant capacity were selected to form the feature vector. A SVM was used as an automatic classifier, obtaining the confusion matrix using leave-one-out cross-validation. Classification performance was assessed with Matthew’s correlation coefficient (MCC) and the AUC_CLASSIFIER_. The most discriminant variables were found to be the total GCL++ thickness (between inner limiting membrane to inner nuclear layer boundaries), evaluated in the peripapillary area and macular retina thickness in the nasal quadrant of the outer and inner rings. Using the SVM classifier, we obtained the following values: MCC = 0.81, sensitivity = 0.89, specificity = 0.92, accuracy = 0.91, and AUC_CLASSIFIER_ = 0.97. Our findings suggest that it is possible to classify control subjects and MS patients without previous optic neuritis by applying machine-learning techniques to study the structural neurodegeneration in the retina.

## 1. Introduction

Multiple sclerosis (MS) causes inflammation, demyelination, axonal degeneration, and neuronal loss in the central nervous system (CNS), hindering axonal conduction and provoking progressive clinical disability in patients.

A single biomarker for diagnosing MS does not exist at present. Oligoclonal bands, magnetic resonance imaging (MRI), and optical coherence tomography (OCT) are all used in clinical practice [1]. MRI is one of the clinical tests that is most widely used in the diagnosis of MS [2]. However, the lesions shown in the images only explain a small fraction of the patient’s clinical symptoms (known as the clinico-radiological paradox). It is, therefore, highly worthwhile to seek new biomarkers capable of determining diagnosis of the disease.

The visual system is an extension of the central nervous system and the neurodegenerative processes of MS frequently manifest in visual pathways [3], making the analysis of them a means of diagnosing the disease and evaluating its progress. OCT uses low-coherence interferometry to produce a two-dimensional image of optical scattering from internal tissue microstructures [4], making it possible to obtain valid information with which to assess the anatomical integrity of the optic nerve and retina. OCT detects the depth at which a difference is present in the refractive index in the transition from one tissue to another. OCT is a rapid (2–3 min) [5], innocuous, cost-effective, and non-invasive test that does not require technical specialization from the practitioner.

The first generation of OCT devices worked in the time domain (time domain OCT:TD-OCT) and depth profiles were acquired by scanning the length of the reference arm to map out the tissue reflectivity [6]. The next-generation technology obtains reflectivity values by taking separate measurements in the Fourier domain for a multitude of wavelengths (Fourier domain OCT:FD-OCT). On a practical level, this is implemented using either a spectrometer and a line scan camera (spectral domain OCT:SD-OCT) or a tuneable swept laser as the light source and a single photodiode detector (swept source OCT:SS-OCT) [6]. This hardware simplification in SS-OCT enables fast scanning speeds of up to 400,000 axial scans per second in non-commercial prototypes [7]. The A-scan speed in commercial SS-OCT devices is 100,000 Hz [8], providing more accurate three-dimensional images of the retina and even the choroid, yielding 8 and 20 μm axial and transverse resolutions in tissue, respectively [9,10]. In non-commercial systems, axial and transverse resolutions of 2 μm over a field of view of 1 × 1 mm^2^ [11] are obtained.

All these characteristics have meant that OCT has revolutionized daily clinical practice in neuro-ophthalmology, because the technique makes it possible to quantify, non-invasively, rapidly, objectively, reliably, and highly reproducibly [12], the major pathological hallmarks of the disease, specifically, inflammation and neuroaxonal degeneration [13]. It also uses the same anatomical area in the follow-up examinations of the eye, which makes it possible to monitor neurological deterioration in neurodegenerative pathologies, thereby serving as a reliable progression biomarker. Some authors consider its reliability to be as high or higher than serial MRI (a test that it is neither innocuous nor cheap, unlike OCT) [14].

The first study using OCT in MS patients was conducted in [15], observing a significant reduction in the retinal nerve fibre layer (RNFL) of patients with MS and previous optic neuritis (ON) when compared with control subjects. Many subsequent studies have confirmed these findings in patients with and without ON, even providing confirmation before clinical symptoms have been presented [16,17,18,19,20].

Two meta-analyses include the latest advances in application of OCT in diagnosis of MS. In the work of Britze et al. [21], it was documented that the thickness of the ganglion cell layer (GCL) was significantly reduced in MS patients both with and without previous ON compared to healthy control subjects. In this meta-analysis (involving a study of 2118 eyes), a reduction in the combined GCL and IPL (inner plexiform layer, together referred to as GCIPL) layers of 6.73 µm was estimated in MS patients without ON. The meta-analysis by Petzold et al. [19] spans 15 studies of retinal thicknesses measured with SD-OCT (conducted in patients without ON, obtaining an estimated peripapillary thickness loss in MS patients and finding that the peripapillary RNFL and macular GCIPL are the most affected layers). In a recent paper [22] studying 97 MS patients without ON episodes, the authors also found, in this case, using SS-OCT, a thinning of retinal thickness in the macular and peripapillary areas in the RNFL and GCL.

Previous studies demonstrate that analysing retinal layers with the latest OCT technology is useful for distinguishing between MS patients and control subjects. None of the papers, however, have employed machine-learning techniques. In this article, the authors go one step further and use machine-learning to optimize the diagnostic capability of the OCT variables and give the technique real clinical applicability, for any population, due to the aforementioned self-learning capabilities.

Machine-learning approaches have been investigated in MS diagnosis. Among these (linear discriminant analysis, random forest, neural networks, etc.), support vector machines (SVM) have demonstrated their effectiveness.

In a classifier, the SVM algorithm maximizes the margin that separates different classes of data. In [23], a SVM acted as a classifier of functional and diffusion MRI data for characterization of relapsing-remitting multiple sclerosis (RRMS) patients. In [24], a SVM was used to predict the course of MS, with demographic and clinical characteristics, MRI features, and characteristics of the first symptom(s) of MS being used as classifier inputs. Zhang et al. [25] diagnosed MS versus controls, comparing three machine-learning-based classifiers, namely, the decision tree, the nearest-neighbour classifier, and the support vector machine.

The objective of this paper is to use machine-learning techniques (a support vector machine) to classify variables obtained with OCT in order to differentiate between control subjects and MS patients. As shown in the general diagram in Figure 1, data were obtained from each patient using an SS-OCT system. In the feature selection stage, the OCT variables with greatest discriminant capacity, evaluated using the area under the receiver operating characteristic (ROC) curve, were selected and used in a classifier implemented in a SVM to obtain the diagnosis.

## 2. Materials and Methods

The study procedures were performed in accordance with the tenets of the Declaration of Helsinki, and the study protocol was approved by the local ethics committees (Clinical Research Ethics Committee of Aragon—CEICA, Zaragoza, Spain). Written informed consent to participate in the study was obtained from all subjects.

Relapsing-remitting MS was diagnosed based on the 2010 revision of the McDonald criteria [26] and was confirmed by a neurologist specializing in MS. Patients with visual acuity less than 0.6 (Snellen scale), intraocular pressure >20 mmHg, a history of optic neuritis, and/or an active MS flare (of any neurological deficit) in the past 6 months preceding their enrolment into the study, or at any of the annual visits, were excluded. Active MS flare was considered a reason for exclusion because acute axonal loss could mask neuronal damage secondary to MS progression (i.e., chronic neurodegeneration), which was the main purpose of this study. Patient disability was analysed using the Expanded Disability Status Scale (EDSS), and the treatment received by each patient was considered.

Axial length was assessed in all individuals. Eyes longer than 25.2 mm and refractive errors ≥5 dioptres (D) of equivalent spherical diameter or ≥3 D of astigmatism were excluded from the study. The participants (MS patients and healthy control subjects) had no concomitant ocular diseases, nor any previous history of retinal pathology, glaucoma, amblyopia, or systemic conditions that could affect the visual system.

Retinal atrophy after acute ON is often more pronounced than the retinal thinning observed in the absence of ON in MS [19]. For this reason, eyes with previous ON were excluded from the analysis.

A complete neuro-ophthalmic examination, including assessment of best-corrected visual acuity using the Snellen chart, contrast sensitivity vision (CSV) using the CSV1000 test, colour vision using the Ishihara test, pupillary reflexes, ocular motility, examinations of the anterior segment, intraocular pressure measurement using the Goldmann applanation tonometer, and papillary morphology, by a fundoscopic exam, was performed on all subjects in order to detect any ocular alteration (such as primary open-angle glaucoma, prior optic neuritis, cataracts, or corneal pathology) that could affect the optic nerve or neuro-retinal structure.

### 2.1. OCT Method

The 3D wide protocol was used for all subjects. This protocol includes a wide scanning range (9 mm high × 12 mm wide) that focuses on both the macula (ETDRS: Early treatment diabetic retinopathy study scan) and the peripapillary area (TSNIT: Temporal-superior-nasal-inferior-temporal scan).

With the ETDRS scan (Figure 2), nine macular areas [21] (which include a central 1-mm circle representing the fovea, and inner and outer rings measuring 3 mm and 6 mm in diameter, respectively, the latter two divided into quadrants by two intersecting lines), central and average thickness, plus macular volume were obtained. The nine areas are denominated as follows: central fovea (CF), inner superior (IS), inner nasal (IN), inner inferior (II), inner temporal (IT), outer superior (OS), outer nasal (ON), outer inferior (OI) and outer temporal (OT).

The diameter of the peripapillary area measured in the TSNIT scan was 3.40 mm. The TSNIT provides measurements of the 4 peripapillary quadrants (superior, nasal, inferior and temporal) and 6 sectors (superonasal, superotemporal, nasal, temporal, inferonasal and inferotemporal, Figure 2).

The ETDRS and TSNIT scans provide separate, automated measurements of different retinal layers, specifically: RNFL (between the inner limiting membrane—ILM and the GCL boundaries), GCL+ (between the RNFL and the inner nuclear layer boundaries), GCL++ (between the ILM and the inner nuclear layer boundaries), and retinal thickness (between the ILM and the retinal pigment epithelium boundaries). Additionally, both the ETDRS and TSNIT protocols provide automated choroidal thickness measurements (from the Bruch membrane to the choroidal-scleral interface).

In this study, the authors recorded the total retinal and choroid thickness in the 9 ETDRS macular areas and the RNFL, retina, GCL+, GCL++, and choroid thickness in the 4 quadrants and 6 sectors of the TSNIT peripapillary area.

All scans were obtained by the same experienced operator and were checked by an experienced rater for segmentation quality immediately after acquisition. If erroneous segmentation was observed, the scan was rejected and repeated. The DRI Triton SS-OCT (1050 nm) provides a quality scale in the image to indicate the signal strength. The quality score ranges from 0 (poor quality) to 100 (excellent quality). Only images with a score >55 were analysed in our study. Poor-quality images were rejected and recaptured prior to data analysis.

In the first phase of the work, the authors obtained the area under the receiver operator characteristic curve (AUC), as to evaluate the discriminant capacity of each of the available variables. Next, the variables with the highest AUC were selected as components of the classifier’s input feature vector.

### 2.2. Support Vector Machine

In a two-class classification problem, a SVM seeks the hyperplane that separates two different classes with maximum margins (support vectors) [27]. If the original data (X) are not linearly separable, a non-linear transformation to a higher dimensional space (H) can be performed using a kernel function, Φ(.):=X→H, such that H improves the separability between the two classes. Kernel function examples include linear, polynomial, radial basis, or Gaussian, etc.

In this paper, the Gaussian function below was used as the transformation kernel.
(1)KGaussian(x,c)=exp(−‖x−c‖2/2.σ2)
where x,c∈ℝP, **c** is the centre of the Gaussian function, and P is the dimension of the feature vector. The σ parameter determines the width of the Gaussian kernel.

### 2.3. Statistical Methods and Classification Assessment

Statistical analyses were performed with the IBM SPSS Statistics 25 software package (IBM Corp., Armonk, NY, USA) and Statgraphics Centurion XVII (Statpoint Technologies Inc., Warrenton, VA, USA). To eliminate possible confusion factors associated with age, exhaustive comparisons were made between the control subjects and the MS patients, analysing the distribution, central trend (means and medians), and variance. A similar comparison was made with the rest of the OCT variables analysed in the study. The comparisons between groups were analysed as follows: Kolmogorov–Smirnov (K–S) tests were used to test distributions, Fisher–Snedecor distribution was used to study the differences in variances, Student’s t distribution was used to analyse the means, and the Mann–Whitney Wilcoxon test was used to compare the medians.

The K–S non-parametric test determines if two samples of data are from the same distribution. The Fisher–Snedecor distribution is a parametric approach to study the equality of the variances of the samples. The analysis of means was performed with the parametric Student’s t test, and the medians were analysed using the Mann–Whitney Wilcoxon non-parametric test. The parametric tests were conducted under normal conditions in the groups.

A p value < 0.05 was considered statistically significant. Correlations were estimated with the Pearson rank correlation. The area under the ROC curve (calculated with SPSS Statistics 25) was employed to assess the discrimination capability of the feature proposed in this study. The AUC quantifies the overlaps between variables distributions. An AUC value of 0.5 implies that the distributions in controls and patients groups overlap. AUC values above 0.9 indicate high diagnostic accuracy [28].

The classifier was implemented using the Classification Learner app included in Statistics and Machine Learning Toolbox 11.6 from MATLAB (Mathworks Inc., Natick, MA, USA). Classification performance was assessed using Matthew’s correlation coefficient (MCC), which summarizes the confusion matrix [29]. MCC ranges are between 1 and –1. A MCC value of 1 denotes a perfect classification, while a value of –1 is a totally erroneous classification, and 0 indicates a random prediction. In a binary classification problem, the MCC is recommended over other parameters like the accuracy and F1 score [30]. The performance of the classifier was evaluated using leave-one-out cross-validation.

## 3. Results

Table 1 shows the results of comparison of the ages of the subjects in the sample. It the compares distributions (Kolmogorov–Smirnov statistic), variances (Fisher–Snedecor distribution), means (Student’s t test) and medians (Mann–Whitney Wilcoxon). The sample can be considered homogeneous in terms of age, as the comparisons made do not reveal any significant differences (p > 0.05). Although there were no differences between groups, there was a larger proportion of females due to the greater incidence of this disease in this gender. Both MS patients and healthy control subjects were of average adult age and were similar. The groups were therefore considered comparable. The time since MS onset in the overall group (male and female) was (mean (standard deviation)): 15.28 (11.17) years. In males it was 15.70 (15.14) years, and in females it was 15.15 (9.97) years. There were no significant differences in the distribution or central trend (means: p = 0.893, medians: p = 0.686), nor in variance (p = 0.079). The EDSS score was 1.55 (0.57). Overall, 20% of the patients were not receiving treatment, 39% were being treated with interferons, and 33% were being treated with immune modulators (principally fingolimod).

Calculating the discriminant capacity of all the variables available in the database (Table 2), it was confirmed that the best results in the classifier (highest MCC value) were obtained by considering the first 3 variables with the highest AUC values as elements of the feature vector, namely, AUC_GCL++_Total_ = 0.879, AUC_ETDRS_IN_Retina_ = 0.859, and AUC_ETDRS_ON_Retina_ = 0.849.

Table 3 shows the values of the three variables used to classify the control group and the patients (GCL++_Total, ETDRS_IN_Retina, ETDRS_ON_Retina), as well as the results of comparing the distributions, variances, means, and medians.

The values of the three variables were higher in the control subjects than in the MS patients, and there are significant differences in the distribution, variance, means, and medians (Table 3). Figure 3a–c shows, in detail, the distribution of the three variables in the control subjects and MS patients. The values in the following subgroups are also shown: C_M: Controls_Male; C_F: Controls_Female; MS_M: MS_Male; MS_F: MS_Female. In both the control group and the MS patient group, the combined GCL++_Total and ETDRS_ON_Retina variables were higher in females than in males.

Figure 3d–f shows the variation with age in the three variables analysed. Of the 12 situations shown in the figures, in our database, there was only a significant correlation with age in the ETDRS_IN_Retina variable (r = 0.414, p = 0.014) in the Controls_Female subgroup (Figure 3e).

The GCL++_Total variable decreases with the years of disease in both males and females, although, in our case, not significantly (p_MALES_ = 0.57, p_FEMALES_ = 0.43), and mostly affecting males. The intercepts of the regression were 130.33 and 134.24 for males and females, respectively, while the slopes were −0.32 and −0.23, respectively (Figure 3g).

The variation in the variable ETDRS_IN_Retina (Figure 3h) with years of disease was negative in males (slope_MALES_ = −0.34) and positive in females (slope_FEMALES_ = 0.27), though, in our case, not significantly (p_MALES_ = 0.52, p_FEMALES_ = 0.67). The variable ETDRS_ON_Retina decreased in both sexes, though, again, not significantly (p_MALES_ = 0.58, p_FEMALE_ = 0.78), with the variation being more pronounced in males (slope_MALES_ = −0.26, slope_FEMALES_ = −0.08, Figure 3i).

In the healthy subjects, we observed a significant positive correlation between ETDRS_IN_Retina vs. ETDRS_ON_Retina (r = 0.674, p = 0.000) and ETDRS_ON_Retina vs. GCL++_Total (r = 0.581, p = 0.000, Figure 4a). In patients, the strength of these correlations increases (Figure 4b): ETDRS_IN_Retina vs. ETDRS_ON_Retina (r = 0.781, p = 0.000) and ETDRS_ON_Retina vs. GCL++_Total (r = 0.769, p = 0.000). In addition, the correlation between ETDRS_IN_Retina vs. GCL++_Total (r = 0.704, p = 0.000) is significant. Our results suggest that the thickness of the outer and inner nasal retina correlate with the total GCL++ thickness, especially in patients with MS without previous episodes of ON and, therefore, it can be assumed that by analysing the thickness of the nasal sector of the retina (ETDRS protocol) it is possible to ascertain the thickness of the peripapillary GCL++ layer. This can be useful if OCT devices with segmentation capability are not available, since even analysing the inner nasal retina with normal OCT devices provides an idea of how the GCL++ layer is behaving.

### Automatic Classifier

Classification was performed by taking the three variables as the inputs and the subject status (control, MS) as the output, using a SVM with a Gaussian quadratic kernel. The K-fold method (K = 1) was used, with a cross-validation with a leave-one-out procedure.

Table 4 shows the confusion matrix obtained. The following values define the characteristics of this matrix: MCC = 0.81, sensitivity = 0.89, specificity = 0.92 and accuracy = 0.91, with an area under the ROC curve of AUC_CLASSIFIER_ = 0.97 (Figure 5a). Only the males in the sample obtain AUC_CLASSIFIER_M_ = 0.92 (n = 24) (Figure 5b), while only the females obtains AUC_CLASSIFIER_F_ = 0.97 (n = 72) (Figure 5c).

## 4. Discussion

In the latest revision to the McDonald criteria [31] the evidence on which to recommend OCT as a definitive biomarker in fulfilling dissemination in space or in time in support of MS diagnosis is considered insufficient, so finding novel parameters to increase sensitivity and specificity in terms of MS is a high priority [32].

The objective of this paper has been to evaluate the MS diagnostic capability of applying machine-learning techniques to SS-OCT data. It has demonstrated that it is possible to distinguish between MS patients and healthy subjects using SVM and it is possible to do so with a high degree of sensitivity and specificity.

The applications of machine-learning techniques have demonstrated their advantages in MRI-based MS diagnosis. For example, a pattern recognition technique was used in [33] to learn a discriminant function, obtaining a sensitivity of 0.82 and a specificity of 0.86, so as to distinguish between MS patients and control subjects using functional MRI (fMRI). In [23], SVMs were used as classifiers (with diffusion tensor imaging—DTI and fMRI data input) between control subjects and relapsing-remitting MS patients, obtaining an accuracy of 89% ± 2%.

The analysis of OCT data for use in diagnosis of MS is an active line of research [16,20,34,35,36]. In a previous paper [37], the authors used a SD-OCT device to take measurements of the RNFL of 106 patients with MS (29% with previous ON) and 115 control subjects. The 768 points registered during circular peripapillary scan acquisition were grouped to obtain 24 uniformly divided locations (15° per location) that formed the feature vector of a multilayer perceptron trained by the back-propagation algorithm. Here, the area under the ROC curve of the classification was 0.945. The novel features of this paper were that the authors studied patients without ON, the measurements were obtained with more sensitive equipment (SS-OCT), the thickness of the GCL was evaluated at peripapillary level, the total retinal thicknesses were evaluated at a macular level, a study of the discriminant capacity of the measurements obtained was conducted to reduce the feature vector to 3 dimensions, and a SVM was used as classifier.

In the initial stages of the study, the authors analysed the variables obtained from SS-OCT that showed the greatest capacity to discriminate between the control subjects and MS patients: GCL++_Total (global GCL++ thickness evaluated at the peripapillary area), ETDRS_IN_Retina (macular retina thickness in the nasal quadrant of the inner ring), and ETDRS_ON_Retina (macular retina thickness in the nasal quadrant of the outer ring).

It is known that retinal layer thickness decreases in humans from the age of 40 onwards. Our results show that when analysing neuroretinal thickness by sex, healthy males suffer an expected loss of inner and outer nasal retinal thickness with age. In contrast, in healthy females, there is an increase in thickness with age, which may be associated with the macular changes and age-associated degeneration produced in this part of the retina, resulting in the appearance of drusen and cellular waste and the accumulation of liquid occurring more frequently in females [38].

In MS patients, males suffered a more pronounced loss with age than females, which would demonstrate greater neurodegenerative damage, an observation that concurs with those of other authors who have found that RRMS is generally more aggressive in males and that male relapse-onset patients accumulate disability faster than female patients [39,40]. MS is at least two to three times more common in females than in males, suggesting that hormones may also play a significant role in determining susceptibility to MS. There is increasing evidence that oestrogen may impact neuroprotection. Grey matter atrophy is an important correlate to clinical disability in MS. In MS animal models, mice treated with the oestrogen receptor (ER)-α ligand or ER-β ligand showed preservation of cerebellar grey matter and Purkinje cells in MRI and histopathological studies [41].

Males with MS showed a tendency towards greater loss in peripapillary GCL++ in the outer retinal sector and, finally, in the inner retinal sector, exhibiting a pattern of topographic and sequential anterograde degeneration corresponding to the papillomacular bundle. In addition, a direct correlation of statistical significance was demonstrated in MS patients between the GCL++ values and those of the outer nasal retina (strong) (r = 0.769, p = 0.000) and the inner nasal retina (moderate) (r = 0.704, p = 0.000).

This loss is reflected in the anterograde degeneration, producing a reduction in average thickness in the macular area. Consequently, it can be deduced that in its early stages, MS manifests principally in the papillomacular bundle.

In addition, females with MS exhibited an upward trend in peripapillary GCL++ thickness. This may have been a reflection of increased immune infiltration by anterograde axonal spread from the brain, as immune diseases such as MS are known to have a greater incidence in females [42,43].

In regard to the years of disease, similar findings have been made. Males present a greater tendency to lose thickness than females, this being mainly evident in the inner retina in the form of greater loss of thickness when compared with females. This suggests more advanced anterograde degeneration in males after the same number of years of disease progression. This more obvious difference in the inner retina may be due to the combination of a female protective role as well as to alterations in the pigment epithelium of the retina and the macular outer retina associated with age.

Conversely, it was also observed that in the variables in which males suffered a more pronounced loss of thickness than females, the males presented greater thicknesses at earlier ages. In this regard, volumetric brain analyses recorded higher measurements in young/adolescent males, with lower measurements at later ages [44]. This suggests that neuroretinal loss is greater when the initial thickness is greater, as suggested in relation to other neurodegenerative diseases, such as glaucoma [45].

In addition, MS patients of both sexes presented lower thicknesses in the 3 most discriminant variables from early ages when compared with healthy subjects, suggesting that the development of the disease could be detected by minor neuroretinal quantification from childhood.

The values recorded in our study in the variables that comprise the feature vector of the classifier are in general agreement with the findings of other similar papers. In [46], the authors observed with SD-OCT that a decrease in peripapillary RNFL thickness was identified in the temporal quadrant (56.6 µm vs. 67.8 µm), mean macular thickness diminished (280 vs. 287 µm, p < 0.05), and there was a moderate and statistically significant direct correlation between RNFL and mean macular thickness (r = 0.69, p < 0.01) in individuals with MS without previous ON (p < 0.05). Petzold et al. [19] analysed papers evaluating SD-OCT. In total, 15 studies (3154 eyes) have been conducted on patients without ON, obtaining an estimated peripapillary thickness loss in MS patients compared with control subjects of 7.08 μm with a 95% CI (8.65 to 5.52) and finding that the peripapillary RNFL and macular GCIPL are the most affected layers. Our findings with an SS-OCT device, which is more powerful and achieves greater penetration of the layers, are similar. The ability of the SS-OCT to generate thousands of cuts quickly increases the resolution of the image and, therefore, it is more sensitive to the detection of very subtle or early changes. Although this loss of 7.08 μm will naturally depend on the number of years that the patient has had the disease and the aggressiveness of it (e.g., patients with progressive forms present greater loss of peripapillary thickness), this paper endeavours to identify the utility of the OCT thickness variables in the early diagnosis of MS without previous ON, even when that 7-micron loss has not yet occurred. It is in this case that SVMs can optimize certain specific variables’ capacity to discriminate between healthy subjects and recent patients.

In [22], it was concluded, using SS-OCT technology, that in MS patients without ON episodes, significant macular thinning was observed in all ETDRS areas in MS patients, and peripapillary GCL++ measurements were found to be reduced in all sectors (p < 0.001) except the nasal quadrant (p > 0.05). In this study, we have demonstrated that the most appropriate variable for optimizing the MCC parameter is GCL++_Total (AUC_GCL++_Total_ = 0.879). Although GCL+ is the layer that has been shown to offer most specific value for measuring neurodegeneration in MS [21,47], our results found GCL++ to be more discriminatory. This may be a consequence of the fact that obtaining this variable using OCT also obtains RNFL thickness, which, as discussed above, has proven to reflect the neurodegenerative process in MS disease to a significant extent. In addition, after a prior evaluation using the AUC, the authors found that the best macular results in the classification were obtained in the outer and inner ETDRS nasal sectors (AUC_ETDRS_IN_Retina_ = 0.859, AUC_ETDRS_ON_Retina_ = 0.849), which correspond to the papillomacular bundle, and were also correlated in strength, showing a pattern of anterograde neurodegeneration.

There are several reasons why, in this paper, the layers indicated in [19,22] have not been used as a feature vector for the SVM classifier. The meta-analysis conducted by Petzold et al. refers to papers that used SD-OCT devices, and the measurements of retinal layers are different in terms of the Fourier-domain and swept-source technologies [48,49]. In relation to [22], it should be noted that the ultimate objective of this paper has been to obtain the best MCC parameter for classification. For that reason, the feature vector variables selected for the classifier are the combination that maximizes this parameter.

From the point of view of the practical application of the proposed method, the retina measurements were recorded using an SS-OCT device that, in a single take, using the 3D wide protocol, analyses a 12 × 9 mm area of the posterior pole, including the macula (ETDRS scan) and optic nerve head (TSNIT scan). The feature vector is made up of data from the macula and the optic nerve and, therefore, if the positive results produced by the classifier are confirmed in a more extensive study, it would be viable to implement the method set out in this paper in OCT software and receive a proposed diagnosis in real time.

OCT has been shown to be a useful tool for diagnosing and monitoring progressive neuroretinal loss in neurodegenerative pathologies such as Alzheimer’s or Parkinson’s disease, as well as for detecting dynamic changes in neuroinflammatory diseases such as MS. Structural variables obtained using OCT help differential diagnosis in disorders that overlap clinically with MS, such as neuromyelitis optic spectrum disorders (NMOSD), myelin-oligodendrocyte-glycoprotein (MOG) seropositive autoimmunity or Susac syndrome (SuS). However, definitive OCT biomarkers for diagnosing MS in early-onset or difficult cases, or for differentiating between the various MS-like entities, have not yet been found [32]. This study has made it possible to detect the 3 most discriminant variables for use in an early-onset MS SVM classifier that could serve as differential parameters in MS-like pathologies or in difficult cases where MRI is not effective.

## 5. Conclusions

Currently, MS diagnosis is based on clinical and standard neuroimaging symptoms defined by the updated McDonald criteria [31]. However, it is accepted that the information provided by OCT plays a relevant role in ascertaining the degree of axonal damage. Definitive MS diagnosis based on the fulfilment of the McDonald criteria can take many years from the appearance of the first symptoms. Accelerating diagnosis has many benefits for patients, as there are several new disease-modifying treatments (alemtuzumab, cladribine, or ocrelizumab) that, if administered at an early stage, would help halt the progression of MS and the associated neural damage.

In this paper, we have demonstrated that it is possible to classify control subjects and MS patients without previous ON by applying machine-learning techniques to study structural neurodegeneration in the retina. Our findings suggest that including OCT in MS diagnostic criteria could help expedite definitive diagnosis.

This study has several limitations that can be overcome in the future. Firstly, it would be desirable to have a database of patients and control subjects from various centres. Secondly, the possible correlations between the OCT variables analysed and the MRI results were not analysed. If the findings were corroborated, the OCT technique would be a more accessible and cost-efficient biomarker. It would also be beneficial to replicate the study using variables obtained with SD-OCT devices, which are those which are currently used in clinical practice.

In addition, and as future lines of research, applying the methods developed for the structural analysis of the retina in other diseases (e.g., Alzheimer’s or Parkinson’s disease), analysing other layers of the retina, and looking for areas with maximum discriminant capacity that do not follow the morphology defined by the ETDRS or TSNIT grids [50] are suggested.

## Figures and Tables

**Figure 1 sensors-19-05323-f001:**
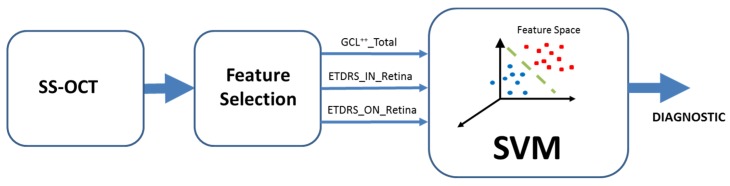
General block diagram. OCT: Optical coherence tomography. SS-OCT: Swept-source OCT; ETDRS: Early treatment diabetic retinopathy study; GCL++_Total: Global GCL++ thickness, evaluated at the peripapillary area between the inner limiting membrane and the inner nuclear layer boundaries; ETDRS_IN_Retina: Macular retina thickness in the nasal quadrant of the inner ring; ETDRS_ON_Retina: Macular retina thickness in the nasal quadrant of the outer ring; SVM: Support vector machine.

**Figure 2 sensors-19-05323-f002:**
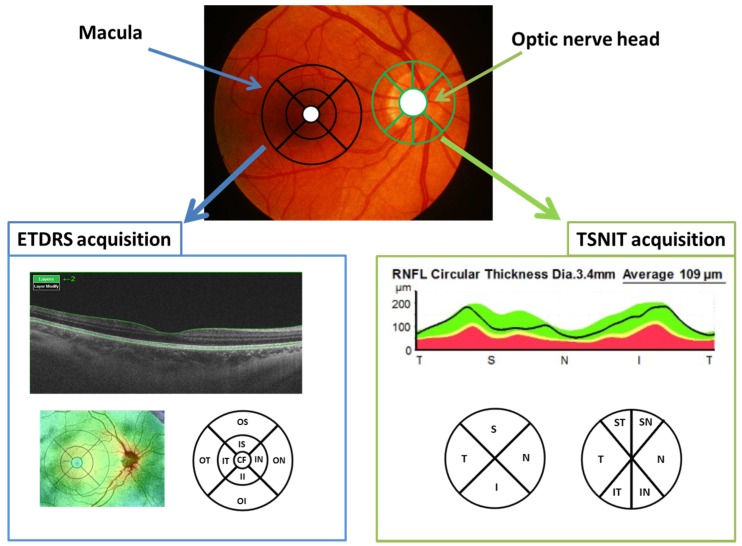
Locations of the OCT scans in the macula and in the optic nerve head. ETDRS: Early treatment diabetic retinopathy study; CF: Central fovea, OT: Outer temporal, OS: Outer superior; ON: Outer nasal; OI: Outer inferior; IT: Inner temporal; IS: Inner superior; IN: Inner nasal; II: Inner inferior, TSNIT: Temporal-superior-nasal-inferior-temporal; ST: Superotemporal; SN: Superonasal; N: Nasal; IN: Inferonasal; IT: Inferotemporal; T: Temporal; RNFL: Retina nerve fibre layer; µm: Micrometres.

**Figure 3 sensors-19-05323-f003:**
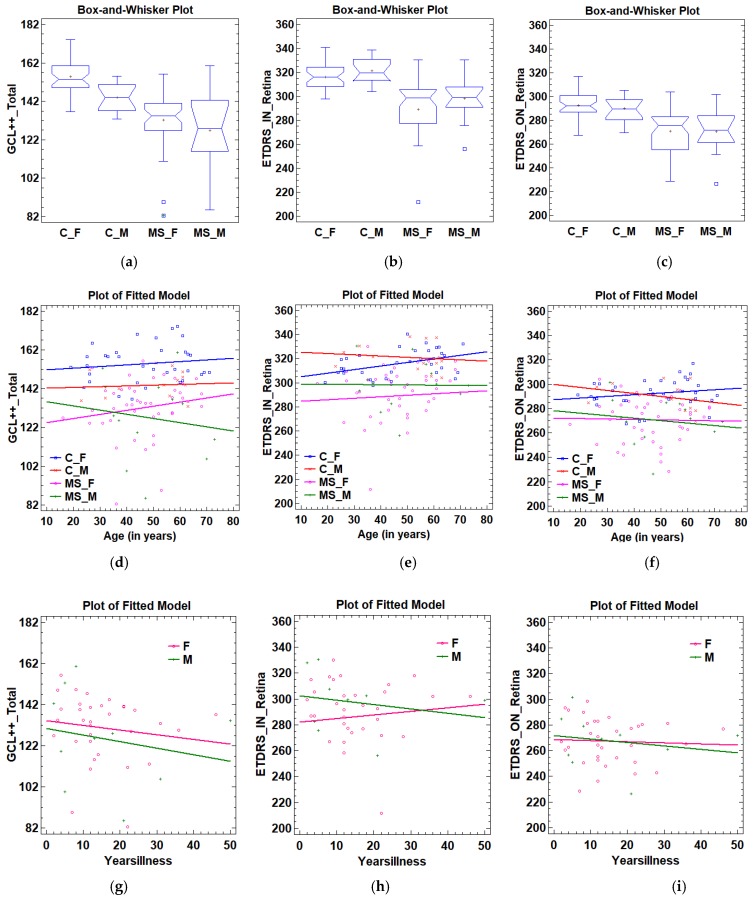
Study of the classifier input variables. (**a**) GCL++_Total according to subjects subtypes. (**b**) ETDRS_IN_Retina according to subjects subtypes. (**c**) ETDRS_ON_Retina according to subjects subtypes. (**d**) GCL++_Total according to age. (**e**) ETDRS_IN_Retina according to age. (**f**) ETDRS_ON_Retina according to age. (**g**) GCL++_Total according to yearsillness. (**h**) ETDRS_IN_Retina according to yearsillness. (**i**) ETDRS_ON_Retina according to yearsillness. C_M: Controls_Male; C_F: Controls_Female; MS_M: MS_Male; MS_F: MS_Female; GCL++_Total: Global GCL++ thickness evaluated at the peripapillary area between the inner limiting membrane and the inner nuclear layer boundaries; ETDRS_IN_Retina: Macular retina thickness in the nasal quadrant of the inner ring; ETDRS_ON_Retina: Macular retina thickness in the nasal quadrant of the outer ring.

**Figure 4 sensors-19-05323-f004:**
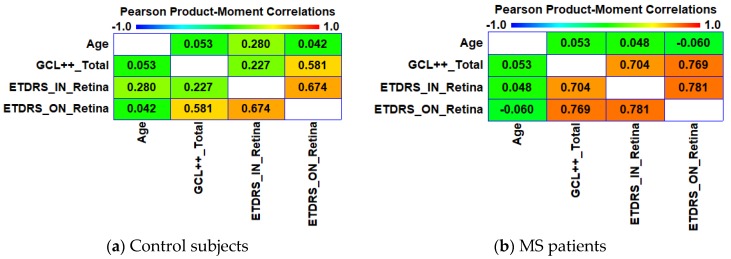
Correlation coefficients between the age and OCT variables. (**a**) In control subjects. (**b**) In MS patients. GCL++_Total: Global GCL++ thickness evaluated at the peripapillary area; ETDRS_IN_Retina: Macular retina thickness in the nasal quadrant of the inner ring; ETDRS_ON_Retina: Macular retina thickness in the nasal quadrant of the outer ring.

**Figure 5 sensors-19-05323-f005:**
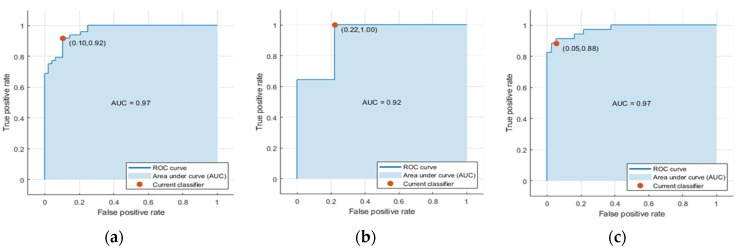
AUC of the classifiers. (**a**) Full sample: AUC_CLASSIFIER_. (**b**) Males: AUC_CLASSIFIER_M_. (**c**) Females: AUC_CLASSIFIER_F_.

**Table 1 sensors-19-05323-t001:** Comparison of subject ages.

	Controls (n = 48)	MS (n = 48)	Test to Compare Distributions	Test to Compare Variances	Test to Compare Means	Test to Compare Medians	AUC (n = 96)	AUC_M_ (n = 24)	AUC_F_ (n = 72)
**GCL++_Total (µm)**	151.65 (10.28)	130.91 (16.63)	K-S = 3.162, p = 0.000	F = 0.381, p = 0.0009	t = 7.759, p = 4.43 × 10^−7^ not assuming equal variances	W = 326.0, p = 3.27 × 10^−11^	0.879	0.750	0.934
**ETDRS_IN_Retina (µm)**	317.52 (11.35)	291.28 (30.71)	K-S = 3.102, p = 8.801 × 10^−9^	F = 0.136, p = 1.29 × 10^−10^	t = 5.937, p = 6.93 × 10^−7^ not assuming equal variances	W = 379.0, p = 3.20 × 10^−10^	0.859	0.845	0.853
**ETDRS_ON_Retina (µm)**	291.72 (11.28)	270.62 (17.96)	K-S = 3.101, p = 8.795 × 10^−9^	F = 0.394, p = 0.001	t = 7.272, p = 0.000 not assuming equal variances	W = 406.5, p = 1.00 × 10^−9^	0.849	0.821	0.859

MS: Multiple sclerosis; SD: Standard deviation; C: Control; vs: Versus; n: Number of subjects; K–S: Kolmogorov—Smirnov statistic, p: significance statistics; F: Fisher–Snedecor; t: Student’s t test; W: Mann–Whitney Wilcoxon.

**Table 2 sensors-19-05323-t002:** Operator characteristic curve (AUC) values obtained in the study.

Area			Retina	Choroid	RNFL	GCL+	GCL++
**ETDRS**		Inner superior (IS)	0.818	0.570	--	--	--
		Inner nasal (IN)	0.859	0.520	--	--	--
		Inner inferior (II)	0.836	0.509	--	--	--
		Inner temporal (IT)	0.812	0.512	--	--	--
		Outer superior (OS)	0.755	0.541	--	--	--
		Outer nasal (ON)	0.849	0.501	--	--	--
		Outer inferior (OI)	0.751	0.512	--	--	--
		Outer temporal (OT)	0.712	0.520	--	--	--
**TSNIT**	**Quadrants**	Temporal (T)	0.805	0.515	0.656	0.82	0.772
		Superior (S)	0.831	0.516	0.832	0.626	0.805
		Nasal (N)	0.733	0.507	0.68	0.685	0.724
		Inferior (I)	0.823	0.52	0.766	0.668	0.805
	**Sectors**	Temporal (T)	0.805	0.515	0.656	0.82	0.772
		Superotemporal (ST)	0.762	0.511	0.742	0.624	0.768
		Superonasal (SN)	0.829	0.502	0.82	0.605	0.829
		Nasal (N)	0.753	0.501	0.704	0.685	0.745
		Inferonasal (IN)	0.769	0.509	0.692	0.679	0.737
		Inferotemporal (IT)	0.770	0.523	0.738	0.596	0.764
	Total		0.835	0.517	0.809	0.76	0.879

ETDRS: Early treatment diabetic retinopathy study; TSNIT: Temporal-superior-nasal-inferior-temporal; RNFL: Retina nerve fibre layer; GCL+ and GCL++: Ganglion cell layers.

**Table 3 sensors-19-05323-t003:** Values of the variables used in the feature vector.

	Controls (n = 48)	MS (n = 48)	Test to Compare Distributions	Test to Compare Variances	Test to Compare Means	Test to Compare Medians	AUC (n = 96)	AUC_M_ (n = 24)	AUC_F_ (n = 72)
**GCL++_Total (µm)**	151.65 (10.28)	130.91 (16.63)	K-S = 3.162, p = 0.000	F = 0.381, p = 0.0009	T = 7.759, p = 4.43 × 10^−7^ not assuming equal variances	W = 326.0, p = 3.27 × 10^−11^	0.879	0.750	0.934
**ETDRS_IN_Retina (µm)**	317.52 (11.35)	291.28 (30.71)	K-S = 3.102, p = 8.801 × 10^−9^	F = 0.136, p = 1.29 × 10^−10^	t = 5.937, p = 6.93 × 10^−7^ not assuming equal variances	W = 379.0, p = 3.20 × 10^−10^	0.859	0.845	0.853
**ETDRS_ON_Retina (µm)**	291.72 (11.28)	270.62 (17.96)	K-S = 3.101, p = 8.795 × 10^−9^	F = 0.394, p = 0.001	t = 7.272, p = 0.000 not assuming equal variances	W = 406.5, p = 1.00 × 10^−9^	0.849	0.821	0.859

MS: Multiple sclerosis; n: Number of subjects; K–S: Kolmogorov–Smirnov statistic, p: Significance statistics; F: Fisher–Snedecor; t: Student’s t test; W: Mann–Whitney Wilcoxon; AUC_M_: Area under the curve for males; AUC_F_: Area under the curve for females; GCL++_Total: Global GCL++ thickness evaluated at the peripapillary area between the inner limiting membrane and the inner nuclear layer boundaries; ETDRS_IN_Retina: Macular retina thickness in the nasal quadrant of the inner ring; ETDRS_ON_Retina: Macular retina thickness in the nasal quadrant of the outer ring; µm: Micrometres.

**Table 4 sensors-19-05323-t004:** Confusion matrix obtained with the Gaussian SVM.

		Predicted Class (Males and Females)		Predicted Class (Males)		Predicted Class (Females)	
		Controls	MS	Controls	MS	Controls	MS
**True Class**	**Controls**	44	4	14	0	30	4
	**MS**	5	43	3	7	3	35

MS: Multiple sclerosis; SVM: Support vector machine.

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
