# Peer review of "Computer-Aided Diagnosis of Multiple Sclerosis Using a Support Vector Machine and Optical Coherence Tomography Features"

_sensors, 2019, doi:10.3390/s19235323_

Round 1
Reviewer 1 Report
The current manuscript reports an interesting work that evaluates the utility of optical coherence tomography (OCT)-derived measures as potential diagnostic biomarkers of multiple sclerosis. For this purpose, MS patients without a history of optic neuritis and demographically matched healthy controls were included and underwent OCT, the data of which were analyzed using an automatic algorithm. Using group comparison, correlation analysis and the area under the curve, the authors report significant variables that could serve as promising biomarkers in MS. This work could add some value to the available literature on this matter. However, some issues need to be addressed.
Statistics :
-It is important to provide a reference for the used software. The authors compared the distribution and dispersion using K-S and Fisher test respectively. Then, they performed both parametric (t test) and non-parametric (Mann Whitney) tests. This needs to be explained.
-Usually, when comparing groups, the normality of distribution and the homogeneity of variance are tested and, afterwards, parametric or nonparametric tests are used accordingly for each variable. An alternative way when some of the data not follow a normal distribution is to employ nonparametric tests for all the variable, especially when the sample size is small.
-It is important to state if the significance of results was corrected for multiple comparisons, or account for this in the discussion.
-It would be also important to mention what are the variables that were included in the correlation analysis.
Results :
-The number of patients and healthy controls seems to be different between the abstract and the results. This needs to be clarified.
-In some circumstances, as in l. 187-188, the variables are mentioned without the unit (i.e., years in this case).
-Some of the important clinical characteristics (disease-modifying therapy, Expanded disability status score) are lacking.
- in figures 3a-c, the authors present independently the data of patients and healthy controls according to sex. If there is a statistical significance of the parameters of interest according to sex, is it important to clarify this in the text and figures, and mention in the statistical analysis the employed test for this subgroup analysis.
- in l. 211-220 and the related figures 3d-f, many of the findings that are mentioned as significant have p values quite above 0.05. The same applies to l. 226-230 where many results are referred to as significant while the p values are above 0.05. This needs to be corrected and taken into consideration in the discussion.
- In some of the figures, the name of parameters are capitalized (e.g., AGE) while others are not (e.g., Years Illness). It would be better to standardize this.
Discussion :
- The discussion needs to account for the proposed changes in the statistical and result sections.
-The obtained variables could significantly discriminate between MS patients and healthy controls which is an interesting finding. What would be clinically relevant is the ability to find parameters that could serve as potential biomarkers and improve the diagnostic accuracy when facing challenging cases. This would allows differentiating difficult MS cased from other neuroinflammatory diseases. This merits to be discussed.
- The statement regarding the 2017 revised McDonald criteria needs to be changed since the cited reference does not seem to recommend OCT and rather states as follows : "despite recognising optic nerve involvement as an important feature of multiple sclerosis, the Panel felt the data concerning the diagnostic sensitivity and specificity of […] optical coherence tomography […] were insufficient to support incorporation into the McDonald criteria at this time. Studies to validate […] optical coherence tomography in fulfilling DIS or DIT in support of a multiple sclerosis diagnosis were identified as a high priority."
- The statement relationship between JCV and MS might be avoided since, based on the actual evidence, JCV screening and monitoring is actually done to prevent the complications of some treatments (e.g., progressive multifocal leukoencephalopathy risk with Natalizumab treatment) and not as biomarker of the disease.
- The limitations of this work need to be highlighted.
Author Response
"Please see the attachment."

Reviewer 2 Report
General
This study develops a machine learning in the diagnosis of MS using SS OCT. The topic is of high interest from the medical point of view, while OCT is a hot topic. The paper is in general well-written, and it can be considered for publication in Sensors, with some necessary improvements, as pointed out bellow.
Specific
1) The English has numerous issues, starting with the Abstract. The entire manuscript has to be revised a lot in this respect.
Please rephrase: in the Abstract “The parameters more of discriminating are” and “Using the SVM classifier obtained the following values”; in Line 62, “clinical symptoms presented”; in Line 69, “Machine-learning approaches have been investigated in” and the entire paragraph; “an SVM acts as”;
2) While the Refs considered are relevant, a few important Refs should still be added, and their content should be reviewed:
- the first paper that introduced OCT in 1991:
Huang, E. A. Swanson, C. P. Lin, J. S. Schuman, W. G. Stinson, W. Chang, M. R. Hee, T. Flotte, K. Gregory, C. A. Puliafito, J. G. Fujimoto, Optical coherence tomography, Science (1991) 254, 1178-1181.
- at least a paper that highlights the advantages of SS (in general, Fourier Domain) OCT vs Time Domain (TD) OCT:
Wojtkowski, M. High-speed optical coherence tomography: basics and applications. Appl Opt 2010, 49, D30-D61.
Drexler, W.; Liu, M.; Kumar, A.; Kamali, T.; Unterhuber, A.; Leitgeb, R.A. Optical coherence tomography today: speed, contrast, and multimodality. J Biomed Opt 2014, 19, 071412.
- state-of-the-art OCT in terms of resolution (equal 2 microns axial and lateral resolution)
Cogliati, A., Canavesi, C., Hayes, A., Tankam, P., Duma, V.-F., Santhanam, A., Thompson, K. P., and Rolland, J. P., MEMS-based handheld scanning probe with pre-shaped input signals for distortion-free images in Gabor-Domain Optical Coherence Microscopy, Opt. Express 24(12), 13365-13374 (2016).
Please also refer to resolution and penetration depth when introducing and discussing OCT.
- a ref on performant SSs (which give that A-scan speed pointed out)
Oh W.Y., Yun S.H., Tearney G.J., Bouma B.E., 115 kHz tuning repetition rate ultrahigh-speed wavelength-swept semiconductor laser, Opt. Letters 30, 3159-3161 (2005).
3) A ref has to be given also on the 2-3 min (acquisition time?) pointed out in the Intro.
4) The type of OCT system (SS vs SD) is not necessarily related to the centre wavelength, please correct in line 55.
5) Line 59: “was conducted by [5],” has to be rephrased in “was conducted in [5],” or “was conducted by Parisi in [5],”. Please correct accordingly wherever such expressions appear.
6) There is no need for subsections in the Intro.
7) The multitude of methods utilized should be commented, at least briefly – for the readers. Right now it looks too much like an enumeration. Please make the paper rigorous, but also attractive for readers.
8) While the statistical analysis performed looks fine, the authors should provide readers with some comments (and interpretation) regarding the results. Right now this part of the manuscript is too much drowned in numbers. Please provide interpretation for the readers – even before the Discussion section.
9) Conclusions are much too brief. Please complete and also provide directions of future work.
In conclusion, this reviewer considers that, as the topic is of interest and the scientific part looks sound, the paper can be considered for publishing, after revisions, in Sensors.
Reviewer 3 Report
This paper extracts retinal thickness features from SS-OCT images of multiple sclerosis (MS) without optic neuritis(ON) and healthy controls, then select three discriminated features (e average GCL thickness (at peripapillary area) and macular thickness at fovea and superior quadrant of the inner ring) for classification by SVM classifier. The main contribution of this manuscript is to find some quantification features to differentiate MSON patients from healthy subject and used as input of SVM to show its effectiveness, where SS-OCT device is used and small of number samples are tested.
There are many works on finding MS biomarkers using OCT imaging, such as the Ref. 9 with SD-OCT and Ref.16. with SS-OCT. The Ref. 9 has concluded that the largest and most robust differences between the eyes of people with multiple sclerosis and control eyes were found in the peripapillary RNFL and macular GCIPL. This paper has the similar findings in lines 155-159, here the average GCL thickness (at peripapillary area) and macular thickness at fovea and superior quadrant of the inner ring have been proved the best results in the final diagnosis. What my main concern is how to prove this. This problem is key for this manuscript and it has to be solved, some comparison experiments should be done. Since all the thicknesses could be obtained from your software in Ref.16, these classification experiments should not be difficult.
Ref. 16 has concluded that “New swept-source technology for OCT devices detects retinal thinning in MS patients, providing increased depth analysis of the choroid in these patients. MS patients present reduced retinal and choroidal thickness in the macular area and reduced peripapillary retinal, RNFL, and GCL thickness.” This manuscript is the further work. All the related thicknesses are calculated in the Ref. 16., the main contribution of this manuscript is to select the three thickness features and analyze them in detail. Please describe the differences between your work and that in Ref.16 to show your concrete contribution. Specifically, Ref. 9 and Ref. 16 both pointed out the important role of RNFL thickness in detecting MS patients. How to show your selected three thickness features performs better?
The study correctly designed and technically sound. However, the data samples are small. MS: 23, Control: 23, where the number of male subjects is 8, and that of female subjects is 36.
The introduction must be improved. There are MS research works using OCT imaging. They are simply mentioned in the discussion. Suggest you review some given significant findings using OCT imaging in the Introduction part and emphasize your new findings in this manuscript.
Line 22: “Twenty asymptomatic of optic neuritis MS patients’ and 22 healthy controls were selected,…”. However, in Section 3, 23 MS and 23 Controls are used in test. Please confirm and modify it.
GCL thicknesses in line 24 is the same as GCL+ thickness in line 146?
Round 2
Reviewer 1 Report
The reviewer would like to thank the authors for the attention they paid to the raised issues. The manuscript greatly improved following revision.
Minor comments remain: The number of patients in the abstract needs to be corrected. Some of the acronyms in the abstract need to be defined upon their first appearance. The manuscript would also benefit of some editing (e.g., lacking spaces)
Reviewer 2 Report
The manuscript has been revised based on all received comments. While it is in a much better shape now, there are still a few (minor) issues that have to be corrected:
1) Please include city and country for Topcon manufacturer of OCT system, including in the Abstract. The same for IBM in 2.3.
2) For the new paragraph on OCT, the statement 'The wavelength of the tuneable swept laser is centred at 1,050 nm, compared to the approximately 840 nm used in SD-OCT [8]' is not true, as there are SS-OCT systems for example that work with 1310 nm wavelengths. In fact, all types of systems, TD-, SD, or SS-OCT can work with one of three center wavelengths: 840, 1050, 1310 nm. This is not dependent on the system, but on the analyzed samples. Tissue contains water that absorb less energy (light) around those wavelengths. For anorganic samples, for example, one could use any wavelength. Please correct.
Attenuation, sensitivity and resolution are another discussion, but such aspects are not necessary to be discussed in detail in this study, although it would be interesting in the Discussion section.
3) Please remove 'advanced' in 'In [25], it is concluded, by using advanced SS-OCT technology'. The system is nor super-speed or ultra-resolution.
4) The two final paragraphs of Discussions, which are the response to Comment 9, are better in the Conclusions section.
In conclusion, this reviewer considers that the paper can be considered in the present form for publication, with the above corrections.
Reviewer 3 Report
The revised manuscript has been improved greatly. The number of test samples is increased. Experiments are redone and three new feature variables are selected (the parameter ETDRS_center is replaced by ETDRS_ON_Retina in the revised version) so that their combination is used to discriminate between healthy subjects and MS patients by SVM classifier based on SS-OCT device.
Many variables are considered in this paper, and how to select variables with high discriminant capacity is a key point. In page 7 (below Table 1), the selection variable rule is given, saying that “Calculating the discriminant capacity of all the variables available in the database obtains the following maximum AUC values: AUCGCL++_Total = 0.879, AUCETDRS_IN_Retina = 0.859, AUCETDRS_ON_Retina = 0.849 and AUCTSNIT_Total_Retina = 0.835.” This is an important point in the proposed method. Please describe it more clearly, including what about “the database”? How to calculate AUC values? List all the AUC values you calculated instead of just the four vales.
I have a concern as follows. In the cover letter, it pointed out that “The appropriate feature vector (which in our case maximizes the MCC parameter) is defined by a number P of parameters (in our study P = 3) and due to the possible correlations existing between these parameters, it is not necessarily convenient to use parameters with the maximum value of AUC. As an example, it can happen that AUCparameter1 = 0.95 and AUCparameter2 = 0.90, but if both have a high correlation, the discriminant information they provide…”. This shows that the authors would like to maximizing the MCC parameter, but they select variables with maximized AUC values. How to ensure to maximize MCC parameter by selecting variables with maximized AUC values?
Some minor modifications are as follows:
1) It would be better to mark the locations of the 9 areas (CF, …, OT) for macular part in figure 2.
